# Dental Caries and Oral Health Status of Psychoactive Substance Abusers

**DOI:** 10.3390/ijerph19105818

**Published:** 2022-05-10

**Authors:** Rashmi Bhavsar, Vandana Shah, Namratha A. Ajith, Kinjal Shah, Ahmed Al-amoudi, Hammam Ahmed Bahammam, Sarah Ahmed Bahammam, Bassam Zidane, Nassreen Hassan Mohammad Albar, Shilpa Bhandi, A. Thirumal Raj, Shankargouda Patil

**Affiliations:** 1Department of Oral Pathology and Microbiology, KM Shah Dental College and Hospital, Sumandeep Vidyapeeth Deemed to be University, Piparia, Waghodia, Vadodara 391760, India; rbhavasar99@gmail.com (R.B.); hod_oralpathology@sumandeepvidyapeethdu.edu.in (V.S.); 2KM Shah Dental College and Hospital, Sumandeep Vidyapeeth Deemed to be University, Piparia, Waghodia, Vadodara 391760, India; naacleo2020@gmail.com; 3Department of Prosthodontics, Faculty of Dental Science, Dharmsinh Desai University, Nadiad 387001, India; kinjalbipinchandra@gmail.com; 4Oral Biology Department, Faculty of Dentistry, King Abdulaziz University, Jeddah 21589, Saudi Arabia; ahmalamoudi@kau.edu.sa; 5Department of Pediatric Dentistry, Faculty of Dentistry, King Abdulaziz University, Jeddah 21589, Saudi Arabia; habahammam@kau.edu.sa; 6Department of Pediatric Dentistry and Orthodontics, College of Dentistry, Taibah University, Medina 46526, Saudi Arabia; sbahammam@taibahu.edu.sa; 7Department of Restorative Dentistry, Faculty of Dentistry, King Abdulaziz University, Jeddah 21589, Saudi Arabia; bzidane@kau.edu.sa; 8Department of Restorative Dentistry, College of Dentistry, Jazan University, Jazan 45412, Saudi Arabia; nalbar01@gmail.com (N.H.M.A.); shilpa.bhandi@gmail.com (S.B.); 9Department of Oral Pathology and Microbiology, Sri Venkateswara Dental College and Hospital, Chennai 600130, India; thirumalraj666@gmail.com; 10Department of Maxillofacial Surgery and Diagnostic Sciences, Division of Oral Pathology, College of Dentistry, Jazan University, Jazan 45412, Saudi Arabia

**Keywords:** dental caries, DMFT, oral health, psychoactive substance abuse, tobacco

## Abstract

Substance-abuse disorders are universally associated with comorbid illness. Tobacco is a widely abused substance across the globe and presents a critical public health problem. The precise correlation between tobacco use and dental caries remains unclear. Thus, the present study aimed to evaluate the correlation between tobacco use and dental caries. Methodology: Based on selection criteria, a total of 270 (age 20–50 years) participants were included in the study, and were categorized as group A (n = 135), consisting of tobacco users, and group B (n = 135), comprising healthy controls (non-users). The Decayed, Missing, and Filled index (DMFT) was used to measure caries status. The Simplified Oral Hygiene index was used to evaluate oral health. Results: The tobacco group reported the use of cigarettes; smokeless tobacco in indigenous forms, such as gutka (areca nut, tobacco, and slaked lime), betel nut chewing; and a combination. Individuals with tobacco habits had a higher prevalence of dental caries (Mean DMFT 4.73 ± 4.32) compared to the non-habit group (Mean DMFT 3.17 ± 3.11 (*p* = 0.001). The Oral Hygiene index was significantly higher (indicating bad/poor oral hygiene) in tobacco abusers than those of non-users (*p* = 0.0001). Duration and frequency of tobacco use were correlated with the levels of moderate and severe caries (*p* = 0.001). Conclusion: Psychoactive substance abuse, such as smoking/smokeless tobacco consumption, is associated with higher prevalence of dental caries.

## 1. Introduction

Psychoactive substance (PS) abuse is a disorder with a cluster of negative cognitive, behavioral, and physiological symptoms associated with the consumption of mind- and behavior-altering substances for non- therapeutic reasons [1]. Substance abuse is of serious concern, as it harms individuals and society as a whole through premature mortality, lost productivity, and social, financial, familial, psychological, medical, and oral health burdens [2]. Commonly abused substances include opiates (prescription or illicit), alcohol, tobacco, cannabinoids, and hallucinogens. Of these substances, tobacco and nicotine use is of particular concern. Tobacco use is estimated to lead to 8 million deaths a year [3]. Tobacco abuse in both smoking and smokeless form is abundant in Asian countries [4,5]. The Global Adult Tobacco Survey found that there are over 267 million tobacco users in India alone, accounting for 12% of the global consumption of tobacco [6]. The abuse of tobacco may lead to 70% of deaths occurring in low- and middle-income countries, and the mortality rate may reach 10 million by 2030 [7,8]. Tobacco consumption takes many forms, and its health effects are indisputably catastrophic.

Smokeless tobacco products have been a cultural mainstay in Asia for more than a millennium, with their use spreading across the globe [9]. Several tobacco products are indigenous to Asia and specifically India. Tobacco consumption patterns are not monolithic. They are influenced by demographics and sociocultural factors [10]. Popular chewable forms of tobacco include paan masala and gutka. Paan masala is a dehydrated mixture of areca nut, slaked lime, spices, and flavoring agents [11]. Gutka is a similar smokeless tobacco product consisting of crushed areca nut with acacia extract (catechu) combined with paraffin wax and sweet or savory flavorings [12,13]. Commercially available flavored chewing-tobacco flakes include zarda (consumed with slaked lime silver, sandalwood oil, saffron, and herbs) and khaini (fermented cut tobacco leaves mixed with slaked lime) [14]. Some of these smokeless tobacco products are kept in the mouth, held between the cheeks and gums or in the vestibule for long periods of time. Cigarettes, beedis (unfiltered indigenous cigarettes rolled in parched leaves), and hookahs remain popular smoking forms of tobacco [15]. Tobacco consumption is not only detrimental to overall human health but leads to pernicious effects in the oral cavity resulting in gingival and periodontal disease [16,17].

Dental caries is the most prevalent non-communicable disease known to man, affecting more than two billion people across the globe [18,19]. Caries can destroy enamel and dentine over time, ultimately leading to pain, infections, abscesses, and possibly sepsis in advanced cases [20].

Tobacco consumption alters the microbiome composition and diversity in a user. Smokers with low caries showed an increased *S. mutans* count in subgingival plaque, while Lactobacilli were observed to be higher in smokers with moderate to high caries [21]. Nicotine increases biofilm formation and bacterial adherence, suggesting that smoking can raise caries prevalence by fostering increased bacterial growth and metabolism [22]. Sharma et al. reported that smokers had higher caries prevalence and caries severity, with smokeless tobacco users having a higher caries burden [23]. This could be due to the degenerative effect of tobacco on salivary glands, promoting caries [24]. Smokeless tobacco products and their constituents can also interfere with the buffering capacity of saliva, rendering the user vulnerable to dental caries [25]. 

The association between tobacco consumption and dental caries has attracted conflicting interpretations. A higher concentration of thiocyanate, a constituent of tobacco smoke, was believed to have a protective effect against caries [26]. In a survey of twelve states in India, adolescents reported using tobacco for oral care as a toothpaste, dentifrice, tooth powder, or mouthwash, believing that ‘tobacco use was good for the teeth’ [27]. A systematic review by Bendetti et al. reported that a majority of studies examining tobacco consumption with dental caries suffered from poor quality, and therefore did not offer validation of association [28]. A systematic review by Jiang et al. reported a positive correlation between tobacco smoking and increased risk of dental caries. However, the overall representativeness of the studies considered was not robust [29]. 

Several divergent accounts have been proposed linking tobacco to caries. The present study aims to provide further insight into the potential association between psychoactive substance abuse and dental caries. In addition, we hypothesize that increased tobacco consumption is correlated with an higher prevalence of dental caries. Thus, in addition to assessing the association, this study assessed the correlation between duration/frequency of psychoactive substance abuse and the severity of the dental caries. The null hypothesis is that there is no difference in dental caries activity between substance abusers compared to non-users.

## 2. Materials and Methods

The study protocol received approval from the institutional review board of KM Shah Dental College and Hospital (SVIEC/ON/DENT/PHD/15002 dated 31 August 2015). This study adhered to the guidelines of the Declaration of Helsinki. 

The present study recruited participants from the outpatient department of KM Shah Dental College and Hospital. All the patients gave written informed consent to participating in the study. All the participants were screened by the primary investigator and were categorized into two equal groups, Group A and Group B. 

### 2.1. Inclusion Criteria for Group A Patients

Dentate participants of 20–50 years of age.History of using tobacco and/or related substances for a minimum of 5 years.Residents of Piparia village, Vadodara city.Consuming only municipal water.

### 2.2. Exclusion Criteria for Group A Patients

Occasional tobacco abusers.Participants with special health care needs.Participants with systemic disease.Pregnant and adolescent females, and females having systemic diseases or hormonal disturbances.

Group B consisted of age- and gender-matched healthy asymptomatic patients visiting the outpatient department for routine oral health checkups, having no history of tobacco consumption. 

Based upon the above-mentioned selection criteria, a total of 270 participants were included in the study, and were categorized as group A (n = 135) and group B (n = 135). 

A brief case history, including details regarding oral hygiene practices and tobacco habits, such as duration, frequency, the quantity of tobacco, and related product intake, was recorded. 

Duration and frequency of habit were recorded from information supplied by participants based on a questionnaire after screening. It contained questions regarding the duration and frequency of habit (days/years) to ascertain each subject’s history of tobacco abuse. Duration was categorized into periods of 1 to 5 years, 5 to 10 years, 10 to 15 years, 15 to 20 years, and more than 20 years of tobacco substance abuse in either smokeless or smoking form. 

Regarding frequency, the questionnaire asked for details of the amounts of smokeless tobacco chewed per day and the number of cigarettes/beedis smoked per day. The frequency of substance abuse was quantified into frequencies of 1 to 4 times/day, 5 to 8 times/day, 9 to 12 times/day, and 13 to 16 times/day for statistical evaluation. 

Each participant was screened on a dental chair and examined further using natural light and diagnostic instruments. The Simplified Oral Hygiene Index (OHI-S) [21] was used to determine oral hygiene status. The OHI-S index Greene and Vermillion, 1964 [21] consists of a combined Debris Index and Calculus index. It is based on 12 numerical determinations representing the amount of debris or calculus found on the buccal and lingual surfaces of each of three segments of each dental arch. Further, the OHI-S index for each patient was calculated by dividing the total sum by the number of groups. Thus, the OHI-S values, comprising the DI-S and CI-S, ranges from 0 to 6, or Good (0 to 1.2), Fair (1.3 to 3), and Poor (3.1 to 6).

Dental caries were assessed using the DMFT Index (Decayed, Missing, Filled teeth) and its pattern, and participants were categorized into three groups: Group I: No caries group: DMFT < 1; Group II: Moderate caries group: DMFT > 1 and <6; Group III: Severe caries group: DMFT > 6. The time taken for clinical examination and record entry was 10 to 15 min for each participant. All tobacco abusers were informed about the ill effects of tobacco and were further counseled for discontinuation of their habits in the Tobacco Cessation Unit of our Institute. 

### 2.3. Statistical Analysis

Data analysis was performed using SPSS Version 22.0 (IBM Corp., Armonk, NY, USA). The data were statistically analyzed using descriptive statistics, Chi-square *t*-test, ANOVA, and binary regression analysis. Data are presented in tables as percentages. Statistical significance was set at *p* < 0.05. 

## 3. Results

This study had 270 participants. The mean age was 32.53 years, and the majority were in the age range of 20–29. Intraexaminer reliability was 0.93 for intraclass calibration. Almost all participants followed the daily oral-hygiene maintenance practice of brushing teeth once in the morning and once at night. 

Gender distribution, as per DMFT categorization among the total population, revealed that males predominantly showed worse DMFT scores (Table 1). 

In the present study, there was a significant difference in the numbers of male versus female participants in the non-habit group, with a female predominance (*p =* 0.001). There was a significant difference in the numbers of male versus female participants in the tobacco abuser group (*p =* 0.001). In various tobacco substance abusers, there was male predominance, with the exception of betel nut. The group of betel nut abusers had female predominance (Table 2).

Tobacco (padiki) was the most commonly consumed form of smokeless tobacco among the abusers studied, followed by gutka. Tobacco (padiki) abuse was more common in males—who numbered 35 (21.47%)—than in females, of whom there were 17 (15.89%) in the tobacco-abuse group, and the difference was statistically significant (*p =* 0.001). A majority of the participants reported tobacco habits of chewing, betel nut abuse, and smoking. The mixed-habit group included participants with gutka abuse in combination with padiki use and consumption of betel nut. However, among the male participants, gutka abuse was most common, followed by tobacco and smoking (Table 2).

In the present study, the mean DMFT in tobacco abusers was significantly higher than that of non-abusers (*p =* 0.0010) (Table 3 and Table 4). A statistically significant difference was found regarding total decayed teeth, total missing teeth, and total filled teeth among tobacco abusers compared to non-abusers (Table 3). Dental caries and OHI scores showed a statistically significant difference between tobacco users and non-users (*p =* 0.0001).

In the present study, there is no significant difference in mean DMFT between male and female participants of the tobacco abusers group (Table 4). Mean DMFT in tobacco abusers was significantly higher than that of non-abusers (*p =* 0.001). Among the varying smokeless tobacco habits, such as tobacco (padiki), there was no significant difference in mean DMFT to that of the non-user group. Mean DMFT in gutka abusers was higher than that of non-users and was statistically significant (*p =* 0.017). In the smokers’ group, mean DMFT was not significantly different from that of non-smokers. Similarly, there was no statistically significant difference in the mean DMFT of betel nut chewers to that of non-chewers. The mean DMFT of mixed tobacco abusers was not significantly different from that of the non-users group. The mean DMFT of overall tobacco abusers was significantly different from that of the non-abusers group. (*p =* 0.001) (Table 4). 

As the frequency of tobacco and related substance abuse increased, there was a statistically significant increase in dental caries and a weak positive correlation (DMFT) (*p* = 0.15 and *p =* 0.013) (Table 5). Further, in moderate and severe DMFT, duration and frequency of tobacco intake were statically significant and were associated with the pattern of dental caries (smooth surface caries with total decayed and missing teeth as compared with filled teeth in the case of total DMFT) (*p =* 0.001). DMFT score increased with the duration of tobacco substance abuse (*r* = 0.24 and *p =* 0.0001) and it was statistically highly significant, with a weak positive correlation.

Gutka abuse showed a statistically significant weak positive correlation with DMFT (*r* = 0.17 and *p =* 0.005). Dental caries increased with tobacco-padiki abuse, but it was not statistically significant (*r* = 0.05 and *p =* 0.384). 

Binary regression analysis (Table 6) showed the odds ratio for gutka abusers was 0.217, and they were 0.2 times less likely to develop dental caries than those of the non-exposed group; the difference was statistically significant (*p =* 0.001). 

The Simplified Oral hygiene index (OHI -S) was significantly higher, indicating poor oral hygiene status in tobacco abusers than those of non-abusers (*p =* 0.0001) (Table 3).

A higher number of participants had good OHI in the no caries group, and as the grade of DMFT increased, the OHI was found to be poor (Table 7). Dental caries and OHI scores showed a statistically significant difference between tobacco users and non-users. OHI scores were poorer with increased frequency and duration of tobacco abuse as compared to those of non-abusers, and there was a statistically significant difference. 

In the present study, an association between oral hygiene with varying dental caries and gender-wise distribution was evident among all the participants of both the tobacco abusers and non-abuser groups. Among male participants, there was a statistically highly significant association between OHI status and dental caries (*p* = 0.001). A majority of 39 (44.32%) out of 88 male participants had poor OHI scores associated with severe caries. Out of seven male participants with good OHI scores, six participants (85.71%) had no caries. Among the female participants, more females (54) had fair OHI. Among 50 female participants with poor OHI grades, a majority of participants (23, 46.00%) had severe caries. Thus, among females, there was a statistically significant association between OHI and dental caries (*p* = 0.020).

## 4. Discussion

Substance-abuse disorders are characteristically relapsing disorders due to chemical dependency and problematic use of mind-altering stimulants/sedatives. Tobacco and nicotine are frequently associated with dependence and addiction. Although studies have recognized that tobacco use contributes to periodontal disease, its link with dental caries is still tenuous. This study was designed to estimate the strength of association and correlation, if any, between tobacco use and dental caries in a rural population. 

We found that the prevalence of dental caries was 66.33% in our sample. This caries prevalence is higher than the national average of 54.16% [30]. The higher prevalence of dental caries is suggestive that tobacco use may be correlated with the development of dental caries. Our results are consistent with previous reports of higher caries prevalence in tobacco users by Tomar et al. [31] and Aguilar-Zinser et al. [32]. Our results reflect those of Rwenyonyi et al. who reported higher DMFT scores and caries incidence associated with tobacco smoking in a Ugandan population [33]. Hagh et al. reported that smokers showed a higher number of caries with lower levels of salivary secretory IgA, resulting in a reduced immune response against cariogenic bacteria [26]. Chewing tobacco has a percentage of sugar content, which could raise the levels of bacterial metabolism, especially when kept in the oral cavity for long periods [34,35]. A note of caution is necessary with comparisons with earlier research, as minute changes in the sample selection viz. rural/urban, occupation, education, etc. may present as confounding factors. 

The mean age of participants in our study ranged from 20 to 50 years, with an average age of 32.53 years. This is consistent with the sample examined by Rooban et al. (mean age of 38.49 years), with the majority of the participants (24.1%) aged between 36 and 40 years [36]. Sumit Kumar et al. found tobacco and related substance habits to be most prevalent among those 25–35 years of age [37]. Higher DMFT scores in tobacco users are in accord with the findings of Badel et al., who examined caries prevalence in a younger population of smokers [38]. Axelsson et al. reported a similar higher caries burden in a sample of older individuals who smoked, and claimed that smoking was a significant risk indicator for tooth loss [39]. These congruent findings of a high prevalence of dental caries in substance abusers suggest that tobacco and substance abuse plays a role in the development of dental caries and poor oral hygiene. 

The relatively young age group indulging in tobacco abuse in our study may reflect peer pressure, inability to cope with stresses, media exposure to tobacco, and ease of availability of tobacco and related substances for younger people. Certain smokeless tobacco products are deemed culturally and religiously permissive [40]. Targeted tobacco advertising and co-opting of ethnic and cultural symbols along with dubious health claims can attract younger individuals and lead to the initiation of habits [41,42]. Subjective perceptions regarding the appetite- and thirst-suppressing elements of chewing tobacco can lead to wider swaths of society embracing tobacco consumption. This holds especially true in rural communities, where addressing the psychological, cultural, and social dimensions of tobacco consumption may be challenging [43].

An increased DMFT score observed across various types of tobacco abuse suggests that tobacco plays a definitive role in the caries process [32,36,44,45]. The type of tobacco substance abuse influences the severity of dental caries occurrence and oral hygiene status [23,26,46,47,48,49,50]. 

We observed that consumption of gutka was correlated with greater DMFT scores. This finding broadly supports previous studies linking substance abuse with a higher caries burden in adult pavement dwellers [51] and street children [52]. 

In the present study, the DMFT score was high in both smokeless and smoked forms of tobacco as compared to controls. Our findings are contrary to previous studies by Rooban [36] and Moller et al. [53], who reported that continuous tobacco consumption can cause wear on the occlusal surfaces of teeth and a reduction in pits and fissures of teeth, leading to smoother surfaces and consequently lesser caries. The presence of thiocyanate in smokers’ saliva may have had a caries-inhibiting effect, leading to a lower DMFT score in smokers in one study [36]. The high caries experience reported in our study may be due to citric acid and sweeteners present in smokeless forms of tobacco. Various sweeteners and lime (calcium oxide in aqueous form) components in smokeless tobacco can alter the local environment of the oral cavity, altering the surrounding saliva and its properties, such as pH, flow rate, viscous saliva, remineralization properties, etc., leading to caries. 

*Streptococcus mutans* uses sucrose for metabolism, and its byproducts lead to increased bacterial adherence and caries generation. This adherence of *S. mutans* significantly increases in presence of nicotine [54]. Nicotine may also enhance the interspecies effect between *Candida*, *Streptococcus mutans*, and Lactobacilli [55]. Alzayer et al. [38] and Ashkanane et al. [39] found that co-aggregation of microorganisms increases with nicotine. 

The unequal and smaller distribution of each category of substance abuse in tobacco users compared to the non-exposed group could explain the lower odds in our study.

We observed that as Oral Hygiene Index (OHI) scores degraded from Good to Poor in tobacco abusers, there was a parallel increase in DMFT score. The combination of poor oral hygiene and increased DMFT scores could occur because tobacco abuse induces alteration of microflora, including *Candida* [21,56], attributed to the constituents of tobacco-containing nicotine, polycyclic aromatic hydrocarbons, polonium, nitrosodietheinal amine, and nitrosoproline [57]. Nicotine leads to alteration of local immune function through impairment of neutrophil function, and consequently converts the healthy oral microbiome to pathogenic oral flora, creating an environment susceptible to diseased oral health [58]. Salivary factors, such as lower pH, reduced buffering capacity, and increased microbial load of *Streptococci mutans* and Lactobacilli may lead to an increased risk of dental caries in tobacco abusers [59,60,61,62,63].

Tobacco and related substance abuse primarily influences dental caries and oral health status, and determines overall DMFT. Tobacco consumption habits varied significantly among the participants, and the DMFT score was influenced by the type of tobacco, frequency, and duration of abuse.

Detailed information on the current oral hygiene practices of the patients was not elicited. Reasons for missing teeth were not recorded in detail. Proximal caries were not confirmed with radiographs. Dental caries being multifactorial, an association between a single positive factor, such as tobacco abuse, and caries prevalence is difficult to conclude with this sample. Prospective studies of large populations could confirm and validate the findings of our study. Public health interventions at the state, community, and health-care provider levels for smoking cessation can reduce the health burden associated with tobacco consumption. A shift in public health policy toward a tobacco-free society through educational, regulatory, and economic strategies can reduce smoking initiation and tobacco abuse among adults and youth, consequently improving oral health.

## 5. Conclusions

Tobacco habits are correlated with a higher prevalence of dental caries. A higher DMFT score was observed in tobacco users. They also showed evidence of poor oral hygiene, which was associated with a higher caries index. Smokeless forms of tobacco consumption are associated with a greater prevalence of dental caries. The relative contribution of socio-behavioral factors is unclear. Future longitudinal studies with larger samples can shed light on the link between tobacco consumption and caries risk. Dentists are ideally placed to educate and motivate patients to discontinue tobacco habits and to encourage health-promotion behaviors in high-risk populations.

## Figures and Tables

**Table 1 ijerph-19-05818-t001:** Gender distribution as per DMFT categorization among the total population.

Gender	Grade of DMFT						Total	*p*-Value	Chi-Square χ^2^
	No Caries Moderate Caries Severe Caries								
	N	%	N	%	N	%			
Male	52	31.90%	57	34.97%	54	33.13%	163	0.745	
Female	38	35.51%	33	30.84%	36	33.64%	107		4.211
Total	90	33.33%	90	33.33%	90	33.33%	270		

**Table 2 ijerph-19-05818-t002:** Distribution of gender and tobacco abuse.

Type of Habit	Male	Female	Total	*p*-Value
No Habit	58 (35.58%)	77 (71.96%)	135 (50.00%)	0.001
Tobacco (padiki)	35 (21.47%)	17 (15.89%)	52 (13.70%)	
Gutkha	43 (26.38%)	6 (5.61%)	49 (25.19%)	
Smoking	25 (15.34%)	1 (0.93%)	26 (10.74%)	
Betel Nuts	0 (0.00%)	5 (4.67%)	5 (0.37%)	
Mixed	2 (1.23%)	1 (0.93%)	3 (1.11%)	
Total	163	107	270	

**Table 3 ijerph-19-05818-t003:** Distribution of habit and varying DMFT parameters.

Dental Caries Status	Tobacco User		Tobacco Non User		*t* Value	dF	*p*-Value	Mean Difference	Std. Error Difference	95% Confidence Interval of the Difference	
	Mean	SD	Mean	SD						Lower	Upper
Total DMFT	4.73	4.32	3.17	3.11	3.39	243.48	0.0010	1.5556	0.4583	0.6528	2.4583
Total D	2.24	2.05	1.67	1.96	2.37	267.45	0.0190	0.5778	0.2440	0.0974	1.0582
Total Missing	0.58	0.97	0.31	0.83	2.43	262.34	0.0160	0.2667	0.1097	0.0506	0.4828
Total Filled	0.93	1.29	1.47	1.82	−2.81	242.22	0.0050	−0.5407	0.1919	−0.9187	−0.1628
OHI	3.62	1.23	2.64	1.11	6.87	265.18	0.0001	0.9776	0.1423	0.6975	1.2578

**Table 4 ijerph-19-05818-t004:** Dental caries distribution among varying tobacco substance abuse.

	DMFT	*p*
**Gender**		0.184 (NS)
Male	4.20 ± 3.93	
Female	3.57 ± 3.68	
**Habit**		0.001 (HS)
No Habit	3.17 ± 3.11	
Habit Present	4.73 ± 4.32	
**Tobacco (padiki)**		0.424 (NS)
Non-user	3.85 ± 3.73	
Abuser	4.37 ± 4.26	
**Gutka**		0.017 (S)
Non-user	3.64 ± 3.62	
Abuser	5.33 ± 4.50	
**Smoking**		0.676 (NS)
Non-user	3.91 ± 3.82	
Abuser	4.27 ± 4.11	
**Betel Nut**		0.706 (NS)
Non-user	3.93 ± 3.83	
Abuser	4.80 ± 4.76	
**Mixed Habit**		0.772 (NS)
Non-user	3.94 ± 3.83	
Abuser	5.00 ± 5.57	

**Table 5 ijerph-19-05818-t005:** Correlation analysis of DMFT and tobacco substance abuse.

		Frequency	Duration	Tobacco (Padiki)	Gutka	Smoking	Betel Nut	Mixed
Total DMFT	Pearson Correlation	0.15	0.24	0.05	0.17	0.03	0.03	0.03
	*p* value	0.013	0.000	0.384	0.005	0.655	0.617	0.634

**Table 6 ijerph-19-05818-t006:** Odds ratio—binary regression to predict the chances of higher DMFT scores among different abusers.

	Unstandardized Coefficients		Odds Ratio	*t*	*p*-Value
	B	Std. Error			
(Constant)	3.170	0.325		9.748	0.000
Tobacco	1.195	0.617	0.123	1.938	0.054
Gutka	2.156	0.630	0.217	3.421	0.001
Smoking	1.099	0.809	0.085	1.358	0.176
Betel Nut	1.630	1.721	0.057	0.947	0.345
Mixed	1.830	2.206	0.050	0.829	0.408

**Table 7 ijerph-19-05818-t007:** Chi-square association test between grades of DMFT and OHI -S in terms of gender in total sample.

Gender	OHI	Grade of DMFT						Total	*p*-Value
		No Caries	%	Mod Caries	%	Severe Caries	%		
Male	Good	6	85.71%	1	14.29%	0	0.00%	7	0.001
	Fair	34	50.00%	19	27.94%	15	22.06%	68	
	Poor	12	13.64%	37	42.05%	39	44.32%	88	
	Total	52	31.90%	57	34.97%	54	33.13%	163	
Female	Good	2	66.67%	1	33.33%	0	0.00%	3	0.020
	Fair	26	48.15%	15	27.78%	13	24.07%	54	
	Poor	10	20.00%	17	34.00%	23	46.00%	50	
	Total	38	35.51%	33	30.84%	36	33.64%	107	

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
