# Peer review of "Dental Caries and Oral Health Status of Psychoactive Substance Abusers"

_ijerph, 2022, doi:10.3390/ijerph19105818_

Round 1
Reviewer 1 Report
The authors studied dental caries in users of phsycoactive substances who are also smokers or non-smokers and in a control group. A study population between the age group 20-50 consisting of both genders were evaluated. The authors concluded that substance abuse increase caries, and caries was observed more in non smokers than in smokers.
Currently, the manuscript is not clearly presented. The rationale for the study, findings and the take home message is not clear. Need significant improvement in presentation too. Please see specific comments below.
Abstract
Line 44: What is Ghutka? Briefly explain here
Line 50: What is meant by oral hygiene index is high? Indicate whether it is associated with good or bad oral hygiene.
Line 52: What is meant by ‘pattern of dental caries’. Please be more specific
Need more clarity when presenting the results in the discussion.
Introduction:
Line 66: What is meant by compound process? Maybe better to rewrite the sentence
Lines 68-69: This sentence can be removed? There is already many literature in the field identifying the caries etiology that you can use to write a new sentence.
Line 70-74: Repetition from previous sentences. Please leave out unnecessary sentences and present facts. Morbidity? Not sure this is accurate. Please rewrite the sentences clearly.
Lines 64-74: Can be trimmed and rewrite to make it cohesive.
Lines 75-82: Provide a definition for pshycoactive substances with a reference? Also breakdown long sentences into smaller sentences and present clearly. Please perform more literature available on pshycoactive substances. Currently there is overuse of/reliance on reference #1.
Line 84: (PS) should be indicated in the first instance pshycoactive substances are introduced in the introduction.
Line 87-91: Please correct grammar and here and throughout the manuscript
Line 92: Repeated sentence.
Line 92-95: Indicate whether these tobacco substances are used in India specifically and a description, since some names are not common and known by researchers in other countries. Also provide a small description how the smokeless forms are used. Chewing?
There are many previous studies on the topic especially individually on the effects of smoking and/or substance abuse on dental caries. Can you provide more such references, summarize their findings and indicate what is the research gap and what is novel and the rationale for the present study?
Methods:
Line 106: Spell out OPD
How was OHI determined? Provide details/references
Were the participants questions on their oral hygiene practices? Frequency of Tooth brushing/flossing? Why would that not have affected the DMFT scores although they are substance users/smokers etc.
Results:
Table 4: What is meant by absent/present is not clear?
Line 142: use ‘period/full-stop’ at the end of sentences here and throughout manuscript. Need to improve on the writing and presentation.
Line 143: Not clear what is meant by pattern of dental caries
Smokers and betal nut abusers seem to have less dental caries compared to non-abusers. With a large number of literature stating otherwise, how would you explain this result? This needs to be discussed in the discussion presenting opposing findings by other authors.
Line 155: OHI needs to be spelled out in its first use and indicated within brackets (OHI)
Line 154: What does it mean when you say the OHI is high or low/ good or poor? Need to clearly indicate this index in terms of dental caries severity. Currently it is not clear
Table 4: Messed up with line numbers
Table 7: ‘GRADE OF’ space between the words. Please check typos here and other tables, text
Line 220: Authors?
Discussion:
Line 220-231: All this information belong in the introduction.
Line 232: Sentence sounds incomplete. 66.33%.....of the study population?
Line 233: Contradictory to line 131
Line 233-235: This sentence does not make any sense. Mean age depends on the age group of the population selected for the study and this does not add anything to the data. Suggest removing the sentence. Otherwise indicate age group which Rooban et al studied and compare which would make more sense. Need to be clear on presenting the data.
The statement mentioned in the conclusion (Line 56-58) on smokers and non-smokers are not discussed. It is a significant statement to make but there is no explanation in the discussion? Need to elaborate more.
Line 277: Candida : first letter capital and italicized
Line 275-279: Provide references related to these statements.
Line 285-289: References?
Line 291: Streptococcus mutans : and italicize scientific names
Conclusion:
Line 307: remove the statement about mechanisms. Mechanisms are not identified in the present study.
Almost all the details in the conclusion can be included in the discussion. Briefly in the conclusion, only indicate the new findings based on your results/data.
Author Response
Response to reviewer’s comments
|
Reviewer’s comment |
Author’s response |
|
Reviewer 1 |
We are grateful for the time taken by the Reviewer in going through our manuscript and providing us with helpful comments. We thank the reviewer for their keen insights and providing us invaluable feedback tp improve our manuscript. Thank you for your time and patience.
We have made the requested changes
|
|
Line 44: What is Ghutka? Briefly explain here:
|
The line has been changed to Gutka is a similar smokeless tobacco product which is a mixture of crushed areca nut with acacia extract (catechu) combined with paraffin wax and sweet or savory flavorings |
|
Line 50: What is meant by oral hygiene index is high? Indicate whether it is associated with good or bad oral hygiene.
|
The line has been changed to Oral hygiene index was significantly higher (indicating bad/poor oral hygiene) in tobacco abusers
|
|
Line 52: What is meant by ‘pattern of dental caries’. Please be more specific |
The line has been removed
|
|
Need more clarity when presenting the results in the discussion. |
We have taken the reviewer’s suggestion and re-edited the entire manuscript for clarity and conciseness with the help of a native English editor.
|
|
Line 66: What is meant by compound process? Maybe better to rewrite the sentence |
Sentence removed
|
|
Lines 68-69: This sentence can be removed? There is already many literature in the field identifying the caries etiology that you can use to write a new sentence.
|
Sentence removed
|
|
Line 70-74: Repetition from previous sentences. Please leave out unnecessary sentences and present facts. Morbidity? Not sure this is accurate. Please rewrite the sentences clearly.
|
Sentence removed
|
|
Lines 64-74: Can be trimmed and rewrite to make it cohesive.
|
The line has been rewritten and shifted to enable better flow and clarity
Dental caries is the most prevalent non-communicable disease known to man, affecting more than two billion people across the globe [18,19]. Caries can destroy enamel and dentine over time, ultimately leading to pain, infections, abscesses, and possibly sepsis in advanced cases [20]. The precise effect of tobacco consumption on dental caries is a much-debated topic of research.
|
|
Lines 75-82: Provide a definition for pshycoactive substances with a reference? Also breakdown long sentences into smaller sentences and present clearly. Please perform more literature available on pshycoactive substances. Currently there is overuse of/reliance on reference #1.
|
The line has been rewritten to improve clarity
Psychoactive substances (PS) abuse is a disorder with a cluster of negative cognitive, behavioral and physiological symptoms associated with the consumption of mind and behavior-altering substances for other than therapeutic reasons [1]. Substance abuse is of serious concern, as it harms an individual and society as a whole with premature mortality, lost productivity, social, financial, familial, psychological, medical, and oral health burdens [2]. Commonly abused substances include opiates (prescription or illicit), alcohol, tobacco, cannabinoids, and hallucinogens.
An exhaustive review of current literature was done (Line 69-109) |
|
Line 84: (PS) should be indicated in the first instance psychoactive substances are introduced in the introduction.
|
The change has been incorporated.
|
|
Line 87-91: Please correct grammar and here and throughout the manuscript
|
We have taken the reviewer’s suggestion and re-edited the entire manuscript for clarity and conciseness with the help of a native English editor.
|
|
Line 92: Repeated sentence. |
Removed. |
|
Line 92-95: Indicate whether these tobacco substances are used in India specifically and a description, since some names are not common and known by researchers in other countries. Also provide a small description how the smokeless forms are used. Chewing?
|
The lines have been rewritten to describe tobacco use in India with a description.
Line 70-84 Several tobacco products are indigenous to Asia and specifically India. Tobacco consumption patterns are not monolithic. They are influenced by demographics, and sociocultural factors [10]. Popular chewable forms of tobacco include paan masala and gutka. Paan masala is a dehydrated mixture of areca nut with slaked lime, spices, and flavoring agents [11]. Gutka is a similar smokeless tobacco product which is a mixture of crushed areca nut with acacia extract (catechu) combined with paraffin wax and sweet or savory flavorings [12,13]. Commercially available flavored chewing tobacco flakes include zarda (consumed with slaked lime silver, sandalwood oil, saffron, and herbs) and khaini (fermented cut tobacco leaves mixed with slaked lime) [14]. Some of these smokeless tobacco products are kept in the mouth, held between the cheeks and gums, or in the vestibule for long periods of time. Cigarettes, beedis (unfiltered indigenous cigarettes rolled in parched leaves), and hookahs remain popular smoking forms of tobacco [15]. Tobacco consumption is not only detrimental to overall human health but leads to pernicious effects in the oral cavity resulting in gingival and periodontal disease [16,17].
|
|
There are many previous studies on the topic especially individually on the effects of smoking and/or substance abuse on dental caries. Can you provide more such references, summarize their findings and indicate what is the research gap and what is novel and the rationale for the present study?
|
Line 90-109 summarizes the current literature on substance abuse and dental caries.
The precise effect of tobacco consumption on dental caries is a much-debated topic of research. Tobacco consumption alters the microbiome composition and diversity in a user. Smokers with low caries showed an increased S. mutans count in subgingival plaque while Lactobacilli were observed to be higher in smokers with moderate to high caries [21]. Nicotine increases biofilm formation and bacterial adherence, suggesting that smoking can raise caries prevalence by fostering increased bacterial growth and metabolism [22]. Sharma et all reported that smokers had higher caries prevalence and caries severity, with smokeless tobacco users having a higher caries burden [23]. This could be due to the degenerative effect of tobacco on salivary glands, promoting caries [24]. Smokeless tobacco products and their constituents can also interfere with the buffering capacity of saliva, rendering the user vulnerable to dental caries [25]. The relationship between tobacco consumption and the increased risk of dental caries has attracted conflicting interpretations. A higher concentration of thiocyanate, a constituent of tobacco smoke was believed to have a caries protective effect [26]. In a survey of twelve states in India, adolescents reported using tobacco for oral care as a toothpaste, dentifrice, tooth powder, mouthwash believing that ‘tobacco use was good for the teeth’ [27]. A systematic review by Bendetti et al reported that a majority of studies examining tobacco consumption with dental caries suffered from poor quality and therefore did not offer validation of association [28]. A systematic review by Jiang et al reported a positive correlation between tobacco smoking and increased risk of dental caries.
Line 107-113 indicates the research gap
A systematic review by Jiang et al reported a positive correlation between tobacco smoking and increased risk of dental caries. However, the overall representativeness of the studies considered was not good [29]. Several divergent accounts have been proposed linking tobacco to caries. As yet there remains insufficient evidence of any etiological relationship. This cross-sectional study evaluated the evidence linking psychoactive substance abuse to an increased risk of dental caries. |
|
Line 106: Spell out OPD: |
Out Patient Department
|
|
How was OHI determined? Provide details/references:
|
Simplified Oral Hygiene Index (OHI-S) [15]determined oral hygiene status. The OHI-S index Greene & Vermillion, 1964 consisted of combined Debris Index and Calculus index. It is based on 12 numerical determinations representing the amount of debris or calculus found on the buccal and lingual surfaces of each of three segments of each dental arch. Further, the OHI-S index for each patient was calculated by dividing the total sum with the number of groups. Thus, OHI-S value comprised of DI-S& CI-S & it ranged from 0 to 6 which were interpreted as: Good (0 to 1.2), Fair (1.3 to 3) & Poor(3.1 to 6).
|
|
Were the participants questions on their oral hygiene practices? Frequency of Tooth brushing/flossing? Why would that not have affected the DMFT scores although they are substance users/smokers etc.
|
Detailed questions on oral hygiene practices were not noted and this is one of the limitation of study.
|
|
Table 4: What is meant by absent/present is not clear?
|
Details added as absent means no habit and present means Habit present i.e. substance Abuser
|
|
Line 142: use ‘period/full-stop’ at the end of sentences here and throughout manuscript. Need to improve on the writing and presentation |
We have taken the reviewer’s suggestion and re-edited the entire manuscript for clarity and conciseness with the help of a native English editor.
|
|
Line 143: Not clear what is meant by pattern of dental caries
|
Caries while calculating total DMFT, individual Total decayed teeth (occlusal or smooth surface, proximal or both caries) were noted on visual and tactile examination. Radiographic examination was not done to confirm proximal caries. Pattern of caries refers to smooth surface caries with total decayed and missing teeth as compared with filled teeth in case of total DMFT. |
|
Smokers and betal nut abusers seem to have less dental caries compared to non-abusers. With a large number of literature stating otherwise, how would you explain this result? This needs to be discussed in the discussion presenting opposing findings by other authors.
|
The discussion has been entirely rewritten according to the reviewer’s comments.
Line 242-265 discusses the caries prevalence and compares with existing literature We found that the prevalence of dental caries was 66.33% in our sample. This caries prevalence is higher than the national average of 54.16% [30]. The higher prevalence of dental caries is suggestive that tobacco may be a risk factor in the dental caries process. Our results are consistent with previous reports of caries prevalence in tobacco users by Tomar et al [31] and Aguilar-Zinser et al [32]. Our results reflect those of Rwenyonyi et al who reported higher DMFT scores and caries incidence associated with tobacco smoking in a Ugandan population [33]. Hagh et al reported that smokers showed a higher number of caries with lower levels of salivary secretory IgA, resulting in a reduced immune response against cariogenic bacteria [26]. Chewing tobacco has a percentage of sugar content, which could raise the levels of bacterial metabolism, especially when kept in the oral cavity for long periods of time [34,35]. A note of caution is necessary with comparisons with earlier research as minute changes in the sample selection viz. rural/urban, occupation, education, etc. may present as confounding factors. The mean age of participants in our study ranged from 20 to 50 years with an average age of 32.53 years. This is consistent with the sample examined by Rooban et al (mean age of 38.49 years) with the majority of the participants (24.1%) aged between 36-and 40 years [36]. Sumit Kumar et al found tobacco and related substances habit to be most prevalent among 25–35 years of age [37]. Higher DMFT scores in tobacco users are in accord with the findings of Badel et al who examined caries prevalence in a younger population of smokers [38]. Axelsson et al reported a similar higher caries burden in a sample of older individuals who smoked and claimed that smoking was a significant risk indicator for tooth loss [39]. These congruent findings of a high prevalence of dental caries in substance abusers suggest that tobacco and substance abuse plays a role in the development of dental caries and poor oral hygiene.
Line 276-293 discusses the contrasting studies and the reasons for the disparity.
In the present study, the DMFT score was high in both smokeless and smoked forms of tobacco as compared to controls. Our findings are contrary to previous studies by Rooban [36] and Moller et al [54] who reported that tobacco consumption continuously can cause wear of occlusion surface of teeth and reduction of pits and fissures of teeth leading to a smoother surface and consequently lesser caries. The presence of thiocyanate in smokers' saliva may have had caries inhibiting effect, leading to a lower DMFT score in smokers [36]. The high caries experience reported in our study may be due to citric acid, sweeteners present in smokeless forms of tobacco. Various sweeteners and lime (calcium oxide in aqueous form), components in smokeless tobacco can alter the local environment of the oral cavity altering surrounding saliva and its properties such as pH, flow rate, viscous saliva, remineralization properties, etc leading to caries.
|
|
Line 155: OHI needs to be spelled out in its first use and indicated within brackets (OHI)
|
Corrections done.
|
|
Line 154: What does it mean when you say the OHI is high or low/ good or poor? Need to clearly indicate this index in terms of dental caries severity. Currently it is not clear
|
Correction done: Our study noted that, as OHI degraded from good to poor in tobacco abusers, there was an increase in DMFT score & this correlation between DMFT and OHI was positive.
|
|
Table 4: Messed up with line numbers: |
Corrections done.
|
|
Table 7: ‘GRADE OF’ space between the words. Please check typos here and other tables, text
|
Corrections done.
|
|
Line 220: Authors?: |
Lines removed |
|
Line 220-231: All this information belong in the introduction.
|
The section has been rewritten.
|
|
Line 232: Sentence sounds incomplete. 66.33%.....of the study population?
|
The line has been rewritten We found that the prevalence of dental caries was 66.33% in our sample
|
|
Line 233: Contradictory to line 131
|
Corrections done.Mean age found in our study was 32.53 with age range of 20 to 50 years.
|
|
Line 233-235: This sentence does not make any sense. Mean age depends on the age group of the population selected for the study and this does not add anything to the data. Suggest removing the sentence. Otherwise indicate age group which Rooban et al studied and compare which would make more sense. Need to be clear on presenting the data.
|
The line has been rewritten according to the reviewer’s comment. The mean age of participants in our study ranged from 20 to 50 years with an average age of 32.53 years. This is consistent with the sample examined by Rooban et al (mean age of 38.49 years) with the majority of the participants (24.1%) aged between 36-and 40 years [36]. |
|
The statement mentioned in the conclusion (Line 56-58) on smokers and non-smokers are not discussed. It is a significant statement to make but there is no explanation in the discussion? Need to elaborate more.
|
The necessary lines have been added to the discussion. Both the discussion and the conclusion have been rewritten.
|
|
Line 277: Candida : first letter capital and italicized:
|
Corrections done
|
|
Line 275-279: Provide references related to these statements.
|
References added
|
|
Line 285-289: References? Line 291: Streptococcus mutans : and italicize scientific names
|
References added Corrections done
|
|
Conclusion: Line 307: remove the statement about mechanisms. Mechanisms are not identified in the present study.
|
Line removed
|
|
Almost all the details in the conclusion can be included in the discussion. Briefly in the conclusion, only indicate the new findings based on your results/data.
|
Findings added in discussion and Conclusion reframed.
|
|
|
|
|
Reviewer 2 |
Authors’ response |
|
The paper presents very well known fact that patients who use tobacco in any form have poorer oral health than patients who do not use tobacco. In that sense the paper adds no new knowledge.
|
We are grateful for the insights provided by the reviewer.
We have edited the manuscript to highlight the novelty of the research question and how this work builds up on the existing evidence.
Data regarding the current status of consumption of tobacco in Gujarat correlated to the subject’s caries status may be of benefit in directing public health policy and adds to the knowledge base regarding the link between tobacco use and dental caries.
|
|
Odds ratios are misinterpreted. In table 6. all odds ratios are lower than 1, indicating that exposed subjects (i.e. tobacco users) have in fact lower odds of an outcome in question than non-exposed subjects (control group). Moreover, only odds ratio in gutka use is statistically significant.
|
Corrections done
|
|
|
|
|
Reviewer 3 |
Authors’ response |
|
This is an interesting study and could provide useful information for health promotion and tobacco cessation in India. The following needs to be addressed in revision: 1) a comparison of two groups is needed. Are they different?
|
We thank the Reviewer for taking the time to provide helpful suggestions and encouraging comments. As suggested by the reviewer, we have made all revisions in the new draft of our manuscript.
Comparison done for both groups of tobacco user and non-user.
|
|
2) The methods section introduced several outcome variables and some of them have more than two categories? how is binary regression appropriate?
|
Outcome variables are dichotomous & nominal in nature, where caries status was also assessed as present or absent, among cases and control, hence performed binary regression analysis. As there is unequal and small sample distribution of each category of tobacco substance abuse, odds results are less than 1 in exposed individuals than non exposed.
|
|
3) the presentation of results is not well presented. The comparison between the use and non-use of tobacco group in dental caries should be the focus. In the logistic regression table (table 6), were other covariates controlled.
|
Thank you for pointing this out. As per reviwer’s comments, we have rewritten the section to focus on dental caries in tobacco users and controls. Other covariates such as age and socio-demographic details were controlled.
|

Reviewer 2 Report
The paper presents very well known fact that patients who use tobacco in any form have poorer oral health than patients who do not use tobacco. In that sense the paper adds no new knowledge.
Odds ratios are misinterpreted. In table 6. all odds ratios are lower than 1, indicating that exposed subjects (i.e. tobacco users) have in fact lower odds of an outcome in question than non-exposed subjects (control group). Moreover, only odds ratio in gutka use is statistically significant.
Author Response

(The authors gave the same response as above.)

Reviewer 3 Report
This is an interesting study and could provide useful information for health promotion and tobacco cessation in India. The following needs to be addressed in revision:
1) a comparison of two groups is needed. Are they different?
2) The methods section introduced several outcome variables and some of them have more than two categories? how is binary regression appropriate?
3) the presentation of results is not well presented. The comparison between the use and non-use of tobacco group in dental caries should be the focus. In the logistic regression table (table 6), were other covariates controlled.
Author Response

(The authors gave the same response as above.)

Round 2
Reviewer 1 Report
Manuscript is much improved.
Please see some minor corrections suggested.
Line 91: Scientific names eg: Lactobacillus should be italicized. No need to italicize Lactobacilli.
Line 294 and 297: Streptococcus mutans: Scientific named should be italicized. The first letter of the genus name should be Capital and the first letter of the species name should be simple. Lactobacillus is the scientific name, Lactobacilli need not be italicized.
Line 309-310: Streptococcus mutans (Italicized) and Lactobacilli (non-italicized)
Author Response
Reviewer 1:
Manuscript is much improved.
Please see some minor corrections suggested.
Line 91: Scientific names eg: Lactobacillus should be italicized. No need to italicize Lactobacilli.
Corrections done
Line 294 and 297: Streptococcus mutans: Scientific named should be italicized. The first letter of the genus name should be Capital and the first letter of the species name should be simple. Lactobacillus is the scientific name, Lactobacilli need not be italicized.
Corrections done
Line 309-310: Streptococcus mutans (Italicized) and Lactobacilli (non-italicized)
Corrections done
Reviewer 2:
The authors have answered my remarks and improved the manuscript. However, the main remark, (i.e. that the paper does not add any new knowledge) still stays. In my opinion, the overall significance of the paper is low and it does not merit the publication in Q1 journal.
Author Response:
While we appreciate the reviewer's feedback, we respectfully disagree. We think this
study makes a valuable contribution to the field because it adds to the knowledge base regarding the association of dental caries status, substance abuse and oral health of an often overlooked rural population. These results can be extrapolated to similar rural populations across South East Asia where similar tobacco habits are prevalent. Our findings have important implications for developing public health policy combating widespread tobacco use that has risen in the past half-decade.
Reviewer 3:
The measurement of variables is not clear.
Measurement of variables added to methodology section.
The presentation of the results is hard to follow and needs to be consolidated to reflect the focus of the study
Results are reframed.

Reviewer 2 Report
The authors have answered my remarks and improved the manuscript. However, the main remark, (i.e. that the paper does not add any new knowledge) still stays. In my opinion, the overall significance of the paper is low and it does not merit the publication in Q1 journal.
Author Response

(The authors gave the same response as above.)

Reviewer 3 Report
The measurement of variables is not clear.
The presentation of the results is hard to follow and needs to be consolidated to reflect the focus of the study
Author Response

(The authors gave the same response as above.)
